# Snapshot prey spectrum analysis of the phylogenetically early-diverging carnivorous *Utricularia multifida* from *U*. section *Polypompholyx* (Lentibulariaceae)

**Martin Horstmann**[1], **Andreas Fleischmann**[2], **Ralph Tollrian**[1], **Simon Poppinga**[3,4]*

**1** Department of Animal Ecology, Evolution and Biodiversity, Ruhr-University Bochum, Bochum, Germany, **2** Botanische Staatssammlung München, München, Germany, **3** Plant Biomechanics Group and Botanic Garden, University of Freiburg, Freiburg im Breisgau, Germany, **4** Freiburg Materials Research Center, University of Freiburg, Freiburg im Breisgau, Germany

* simon.poppinga@biologie.uni-freiburg.de

## Abstract

*Utricularia multifida* is carnivorous bladderwort from Western Australia and belongs to a phylogenetically early-diverging lineage of the genus. We present a prey spectrum analysis resulting from a snapshot sampling of 17 traps–the first of this species to our knowledge. The most abundant prey groups were Ostracoda, Copepoda, and Cladocera. The genus cf. *Cypretta* (Cyprididae, Ostracoda) was the predominant prey. However, a high variety of other prey organisms with different taxonomic backgrounds was also detected. Our results indicate that *U. multifida* may potentially be specialized in capturing substrate-bound prey. Future approaches should sample plants from different localities to allow for robust comparative analyses.

## Introduction

Carnivorous bladderworts (*Utricularia* spp, Lentibulariaceae, Lamiales) catch their prey with sophisticated suction traps [1–3]. Prey spectra have been thoroughly investigated for several free-floating aquatic species from *U*. section *Utricularia*, revealing that members of Acaridae, Crustaceae (especially Cladocera, Copepoda and Ostracoda), Gastropoda, Nematoda, Rotifera, and Tardigrada are commonly caught [4–13]. Mosquito larvae also fall prey to their bladders traps quite regularly [14,15]. Furthermore, a multitude of 'algae' (diatoms, Chlorophyceae, etc.), ciliates, bacteria and protozoa can be found inside the traps and may be part of complex food webs [11,16–23].

For those bladderwort species that are not freely floating in water but are affixed to the substrate (i.e., submerged or emersed terrestrials, including lithophytes, epiphytes, and rheophytic species), only little information regarding the prey spectra exists. Acaridae, Crustacea, and Rhizopoda were found in herbarium material [4], members of Adenophorea, Branchiopoda, Chelicerata, Eutardigrada, Insecta, Maxillopoda, and Ostracoda were found in traps of *U. uliginosa* [24], and metazoa such as gastrotrichs, nematodes and rotifers, as well as protozoa such

**Data Availability Statement:** All relevant data are within the paper and its Supporting Information files.

**Funding:** SP gratefully acknowledges the financial support by the Academic Research Alliance JONAS ("Joint Research Network on Advanced Materials and Systems"). The article processing charge was funded by the Baden-Württemberg Ministry of Science, Research and Art and the University of Freiburg in the funding programme Open Access Publishing. The funders had no role in study design, data collection and analysis, decision to publish, or preparation of the manuscript.

**Competing interests:** The authors have declared that no competing interests exist.

as *Vorticella* spp. (Ciliophora) and numerous algae (especially *Frustulia* sp.) were found in the traps of *U. volubilis* [25].

The phylogenetically early-branching *U. multifida* from *U.* section *Polypompholyx* is an affixed submersed species from the south west corner of Western Australia [26] (Fig 1). In contrast to the typically lentiform, more or less thin-walled, and frontally accessible traps of most other species, *U. multifida* (and two close allies) has thick-walled traps, which are triangular in a transverse section, and an entrance region which can only be accessed from lateral sides [27–29]. This species has drawn some interest recently due to the fact that suction could not be observed in traps during laboratory experiments [30–32], although earlier investigations by Lloyd [28] state that its traps are indeed capable of suction. It was consequently theorized that *U. multifida* may possess an exceptional non-motile trap type similar to the eel trap type found in closely related *Genlisea* corkscrew plants [33], which allow easy entry but prevent exit of prey by structural obstacles. However, no reports on the spectrum of naturally caught prey as well as on the actual process of prey capture are available so far. To gain first insights into the diet and possible prey preference of this enigmatic species, we performed a snapshot prey spectrum analysis on traps collected in the habitat.

## Materials and methods

Two plants with 17 filled traps were collected on 04.10.2008 at a permanently wet seepage site near Marbellup (Western Australia), growing on oligotrophic quartzitic sand-peat soil covered by a ca. 0.5–1.0 cm, slowly flowing water film (Fig 1). Plant material was fixed in aceto-ethanol (3:1) (cf. [9,23]).

Traps were opened with forceps at the Department of Animal Ecology, Evolution and Biodiversity of the Ruhr-University Bochum, Germany and prey was carefully rinsed out. Block bowls were used and covered with a glass plate as often as possible to keep evaporation low (and thus evaporation-induced convection in the samples). Prey was separated from detritus in several portions, presorted into groups with the help of eyelashes, photographed with an Olympus SZX 16 (Olympus, Tokio, Japan) stereo microscope equipped with a TSO camera (Thalheim Spezialoptik GmbH, Pulsnitz, Germany), counted and identified according to the literature [34–41]. Trap sizes were measured with the same optical setup.

## Results

In total, 233 prey items were found inside the 17 traps investigated (Table 1). Due to the varying degrees of digestion, the identification down to the genus level was not possible for many items. The most common prey groups were Ostracoda (112 items), Copepoda (80), and Cladocera (34) (Fig 2). The by far most abundant identifiable prey genus was cf. *Cypretta* from the ostracod family Cyprididae (107 items) (Fig 3A and 3B). Furthermore, members of the copepod superfamilies Cyclopoida (38) (Fig 3C) and Harpacticoida (10) as well as numerous not further identifiable copepods (32) constituted similarly abundant prey groups. The cladoceran genus *Macrothrix* (Fig 3D) was also found in comparably large amounts (28).

One taxon of Acari (Hydrachnidia), four members of Insecta, and two further unidentified prey items were also found. From Insecta, members of Coleoptera (cf. Dytiscidae) in larval stadium (Fig 3E) and Diptera-Chironomidae (in one case cf. *Larsia albiceps*) (Fig 3F) occurred in the traps. Algae, as commonly found in the traps of many *Utricularia* species, could not be detected.

Most captured prey had a size between 0.2–0.9 mm. Only a few prey items exceeded this length. The largest intact prey item we found was an ostracod with a body length of 1.2 mm. The total sizes of the captured coleopteran larva and midge larvae can be estimated to be around 1.3–1.8 mm, based on the sizes of the preserved head capsules found inside the traps.

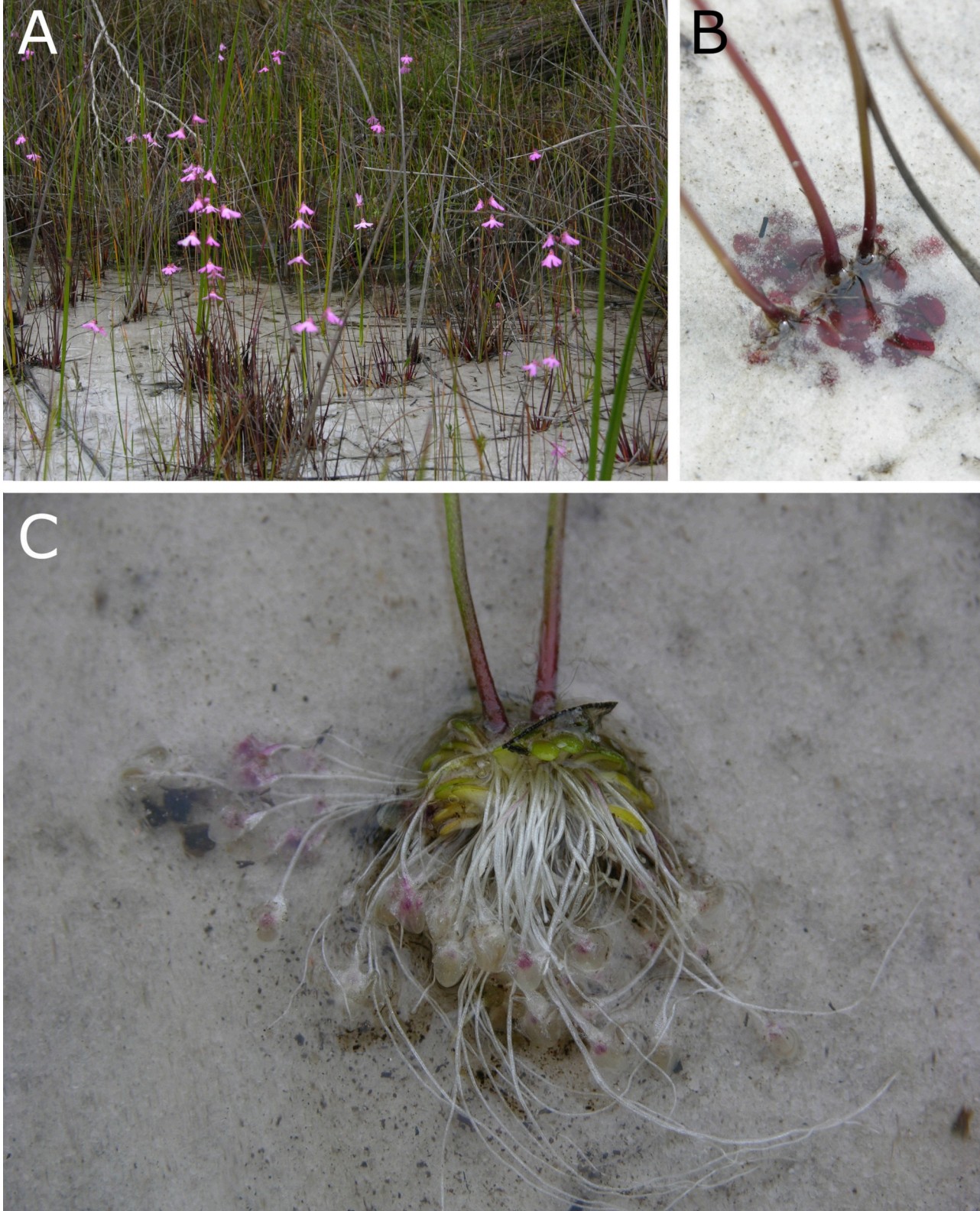

**Fig 1. *Utricularia multifida* in its habitat.** A large population of *U. mutlifida* growing in a permanently wet seepage site near Marbellup (Western Australia), on oligotrophic quartzitic sand-peat soil covered by a ca. 0.5–1.0 cm, slowly flowing water film. (A) A stand of flowering plants. (B) The

plants grow submersed and are partly buried in the soil. (C) An excavated plant. The inflorescence stalks, green photosynthetic leaves, and pale stolons carrying the traps are well visible. Photos by A.F.

With an average trap length of 2.3 mm (min: 1.8 mm; max: 2.7 mm) and width of 1.9 mm (min: 1.5 mm; max: 2.3 mm) (S1 Table), we observed a similar overall size as reported in previous literature [27–32].

## Discussion

Our snapshot analyses of 17 traps from the affixed aquatic *U. multifida* identifies three main crustacean groups as abundant prey from the sampling site, namely Cladocera, Copepoda, and Ostracoda. These groups are also commonly reported as prey of free-floating aquatics [4–12] and, partly, also of non-aquatic species [4,24]. Among all prey items determined, the ostracod genus cf. *Cypretta* was most abundant. It is characterized by its small body size of about 1 mm and the septae inside the carapace valves. It occurs in open, still or slowly flowing waters worldwide with many species, mostly in the southern hemisphere [26,36,39]. *Cypretta* is a grazer foraging for food on substrates, which is typical for most ostracods.

Copepods, which were also abundant in the traps, inhabit soil as well as free fresh water. Especially harpactocoids, which were some of the copepod prey items found in the traps, are mainly substrate-bound species, as they swim poorly [42,43]. Most Harpactocoida inhabit sandy interstitial habitats as found on the sampling site.

The identified trapped Cladocera species are mainly members of the family Macrothricidae. This family barely practices free swimming and moves forward with small leaps, by crawling with the limbs or using its antennas as levers. The long spines of the antennae on the endopods are also used for burrowing in the substrate to search for food [35,44].

The other taxonomic groups found inside the *U. multifida* traps can be regarded as typical (but relatively rarely occurring) bladderwort prey: members of Acari as well as larvae of diptera

**Table 1. Recorded prey items.**

| Group | Class/order | Family | Genus | Species epitheton | Numbers |
|---|---|---|---|---|---|
| Acari | Hydrachnidia | | | | 1 |
| Crustacea | Onychura | Chydoridae | *Saycia* | cf. *cooki* | 1 |
| Crustacea | Onychura | Macrothricidae | *Echinisca* | | 3 |
| Crustacea | Onychura | Macrothricidae | *Macrothrix* | | 28 |
| Crustacea | Onychura | Podonidae | | | 1 |
| Crustacea | Onychura | | | | 1 |
| Crustacea | Copepoda/Cyclopoida | | | | 38 |
| Crustacea | Copepoda/Harpacticoida | | | | 10 |
| Crustacea | Copepoda | | | | 32 |
| Insecta | Coleoptera | cf. Dytiscidae | | | 1 |
| Insecta | Diptera | Chironomidae | cf. *Larsia* | cf. *albiceps* | 1 |
| Insecta | Diptera | Chironomidae | | | 1 |
| Insecta | | | | | 1 |
| Crustacea | Ostracoda/Cypridoidea | Cyprididae | cf. *Cypretta* | | 107 |
| Crustacea | Ostracoda/Cypridoidea | | | | 5 |
| unidentified | | | | | 2 |
| | | | | **Total number of prey items:** | **233** |

Taxonomic background and numbers of the in total 233 prey specimen found in the traps of *Utricularia multifida*.

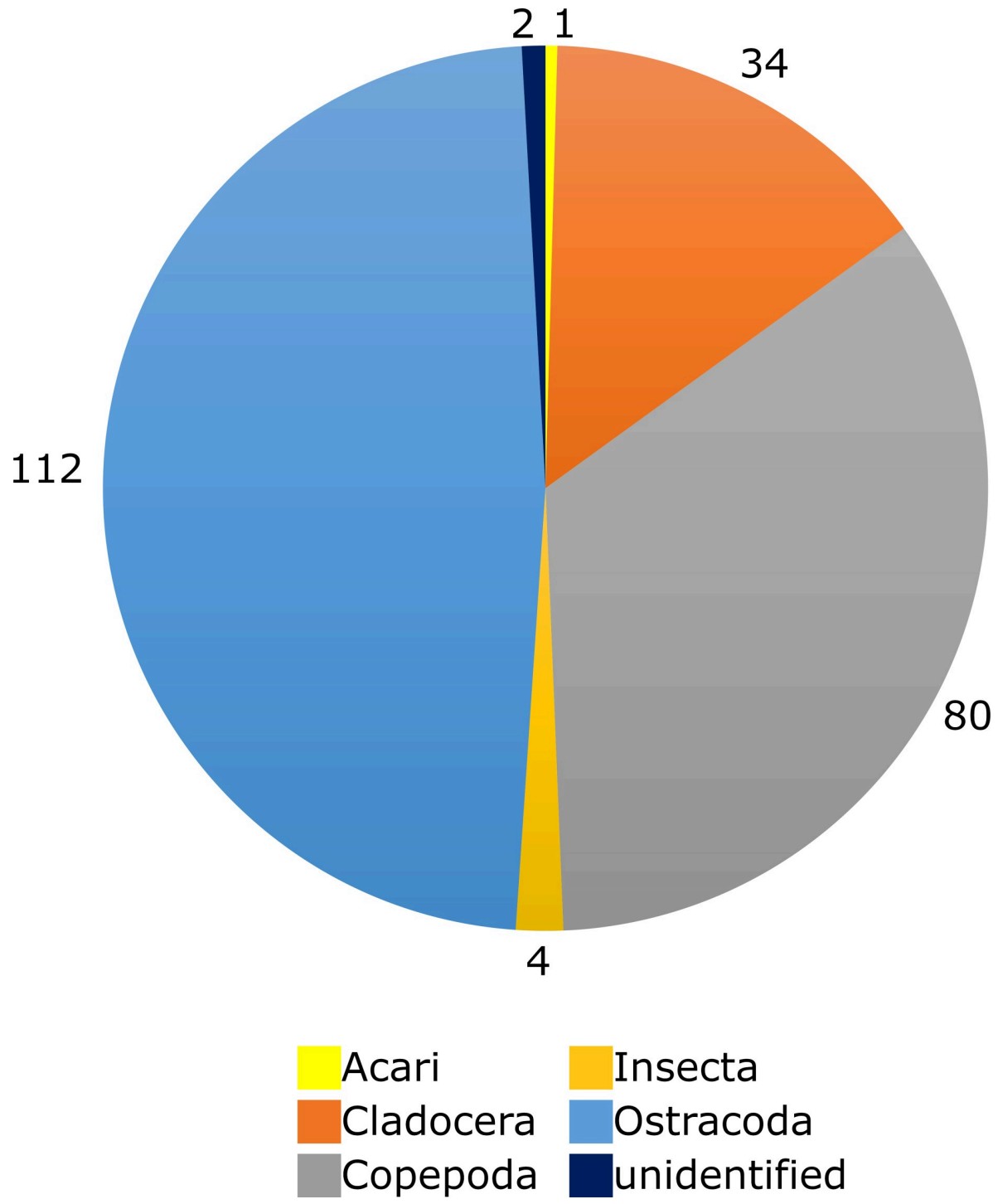

**Fig 2. Abundancy of prey groups from the *U. multifida* traps investigated, depicted as pie chart.** The total numbers of prey items found is indicated for each group. With 112 found prey items, Ostracoda (blue) represents the main prey group, followed by Copepoda (grey, 80 items) and Cladocera (red, 34 items). Insecta (orange, 4 items) and Acari (yellow, 1 item) are much lesser represented. Two prey items could not be identified (dark blue).

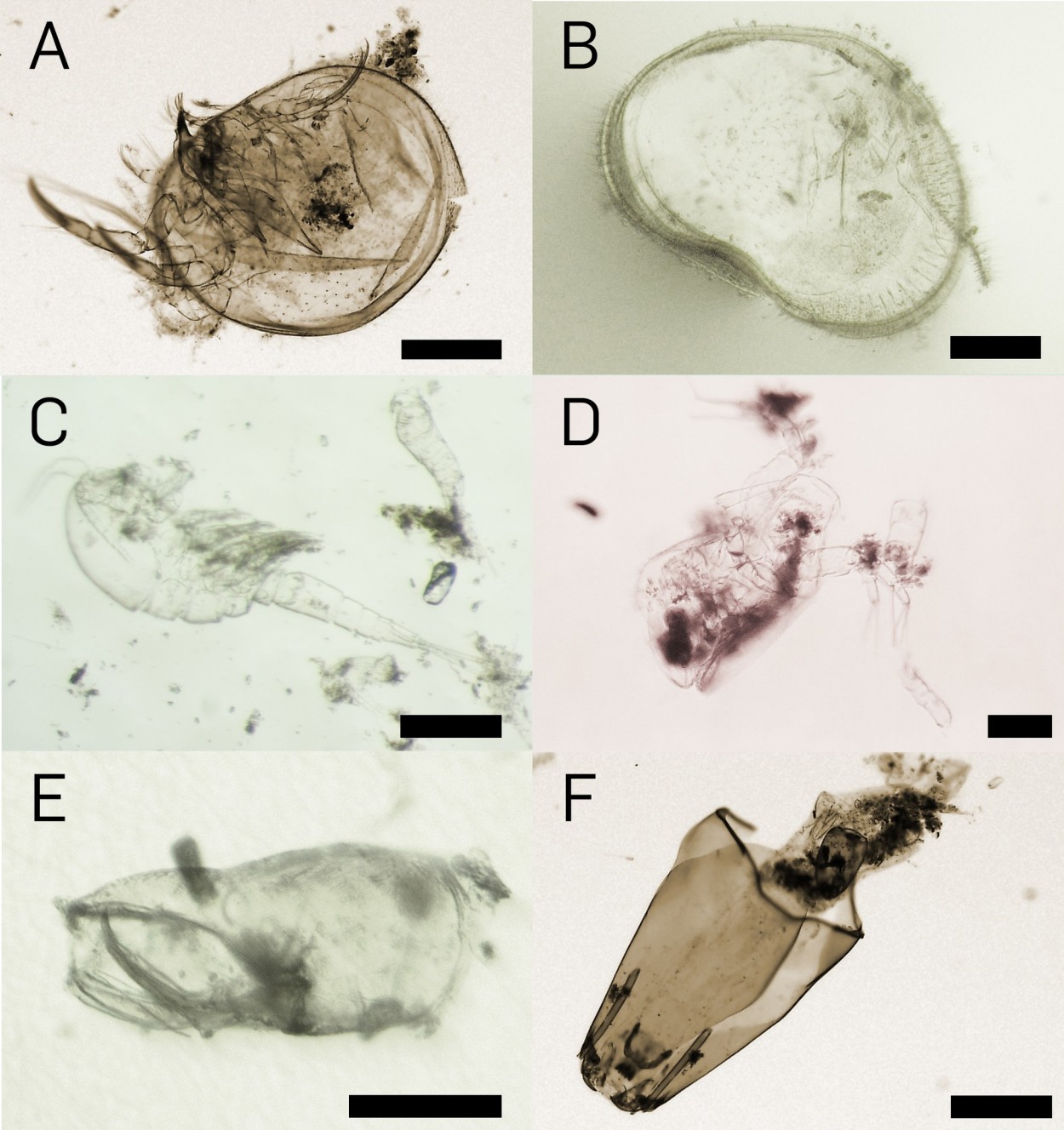

**Fig 3. Prey items detected inside the traps of *U. multifida*.** (A) Ostracoda (cf. *Cypretta*), whole animal, scale bar = 250 μm, (B) Ostracoda (cf. *Cypretta*), carapace half with septae, scale bar = 200 μm, (C) Copepoda, Cyclopoida, scale bar = 200 μm, (D) Macrotricidae, *Macrothrix*, 1st antenna dilated distally, scale bar = 100 μm, (E) Dytiscidae, head of larva, scale bar = 100 μm, (F) Chironomidae, cf. *Larsia*, scale bar = 250 μm.

and water beetles (Dytiscidae) have also been reported from the traps of aquatic species [7,14,18,45,46]. These prey species are characterized by free swimming or substrate-bound behavior.

In summary, our snapshot prey analysis as presented here reveals that *U. multifida* is able to capture prey of a wide morphological and taxonomical range, with substrate-bound crustacean

prey being prevalent. It is therefore imaginable that *U. multifida* is specialized on capturing such prey, which crawls towards the trap entrance zones and becomes captured. However, the prey spectrum may strongly depend on the locality where material was sampled, as all prey groups determined in this study are typical for small, shallow and temporary ponds in general. Especially crustacean populations can rise exponentially after flooding and highly abundant species may be overrepresented in the traps accordingly. Comparative analyses from several sites are necessary to investigate this further, as recently performed in the Droseraceae for the aquatic waterwheel plant (*Aldrovanda vesiculosa*) [47] and for terrestrial annual sundews from *Drosera* sect. *Arachnopus* [48]. Due to the highly ephemeral nature of *U. multifida*, comparative analysis during different times of the year are not feasible.

Since knowledge on the actual plant (predator)–prey interaction is completely absent, future studies should concern also about the actual behavior of prey (esp. *Cypretta*) in the vicinity of the traps and how it is actually caught (cf. [2]). Does prey get sucked into the trap, or does it actively crawl inside? The darkish traps of *U. multifida* are probably highly attractive for substrate-bound, shelter-seeking prey (as reported in this study), which may try to enter the trap and thereby become captured. Our observation that no algae were inside the traps, which are otherwise commonly to be found in motile bladderwort suction traps [11,18,19,25], also hints towards the existence of a passive trap type. However, we are certainly aware that much more detailed physiological, biomechanical and functional-morphological analyses are required to investigate this further. Not only in the context of this current debate it is important to continue research on the functional principle of the traps of *Utricularia* (and especially on *U. multifida*) and on its prey (cf. [1–3,28,31,49]).

## Supporting information

**S1 Table. *Utricularia multifida* trap length and width measurements.**
(PDF)

## Acknowledgments

AF thanks Allen Lowrie (Duncraig, Australia) for helpful correspondence over many years and for help in acquiring plant material in 2008.

## Author Contributions

**Conceptualization:** Andreas Fleischmann, Simon Poppinga.

**Investigation:** Martin Horstmann, Simon Poppinga.

**Methodology:** Martin Horstmann.

**Resources:** Andreas Fleischmann, Ralph Tollrian.

**Writing – original draft:** Martin Horstmann, Simon Poppinga.

**Writing – review & editing:** Andreas Fleischmann, Ralph Tollrian.

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
