## [Decision Letter · Decision Letter 0]

17 Mar 2021

PONE-D-21-00372

Snapshot prey spectrum analysis of the phylogenetically early-diverging carnivorous Utricularia multifida from U. section Polypompholyx (Lentibulariaceae)

PLOS ONE

Dear Dr. Poppinga,

Thank you for submitting your manuscript to PLOS ONE. After careful consideration, we feel that it has merit but does not fully meet PLOS ONE’s publication criteria as it currently stands. Therefore, we invite you to submit a revised version of the manuscript that addresses all the points raised by the three referees during the review process.

We look forward to receiving your revised manuscript.

Kind regards,

Ofer Ovadia

Academic Editor

PLOS ONE

Journal Requirements:

Reviewers' comments:

Reviewer's Responses to Questions

**Comments to the Author**

1. Is the manuscript technically sound, and do the data support the conclusions?

Reviewer #1: Yes

Reviewer #2: Yes

Reviewer #3: Yes

2. Has the statistical analysis been performed appropriately and rigorously? 

Reviewer #1: Yes

Reviewer #2: No

Reviewer #3: Yes

3. Have the authors made all data underlying the findings in their manuscript fully available?

Reviewer #1: Yes

Reviewer #2: Yes

Reviewer #3: No

4. Is the manuscript presented in an intelligible fashion and written in standard English?

Reviewer #1: Yes

Reviewer #2: Yes

Reviewer #3: Yes

5. Review Comments to the Author

Reviewer #1: This interesting paper by German authors aims at determining the prey spectrum in traps of aquatic Utricularia multifida, an exotic and evolutionally early-diverging lineage of the species endemic to Australia. As there are disputations among different scientific teams concerning the functioning of U. multifida traps (opened passive vs. actively capturing closed traps), this novel study not only describes the field-based prey spectrum of this remarkable species for a comparative purpose with other aquatic Utricularia, but also indirectly determines the trap functioning and, moreover, sheds light on the phylogeny of Utricularia suction traps. In this study, the authors have convincingly proven that field-grown U. multifida does capture its prey of different taxonomical groups like other typical aquatic Utricularia species, but the mechanism of prey capturing stays unclear. I add some comments or questions to improve slightly the manuscript.

p.3, l.58-59: “suction could not be observed in traps during laboratory experiments [29-31],….“ : maybe that it is not so important that they were laboratory experiments but that the studies (at least No. 31 – Plachno et al. 2019) were conducted on TC-raised plants. This fact could be crucial for the type of functioning of U. multifida traps: active in the field but passive in TC. I suggest that the authors mention that e.g.: “ observed in traps of tissue-culture raised plants during laboratory experiments [29-31],……“

p.4, l.79: Could the authors specify what was the approximate size/length of the traps? Range?

p.5 or 7: Could the authors specify the maximal approximate length of the prey captured? This datum might be substantial for decision whether the traps are active (i.e., negative pressure, water suction) or passive (i.e., eel traps).

In conclusion, this short communication should be published after a minor revision.

Reviewer #2: This paper is interesting and contributes to an area of the literature which seems quite understudied, but it is extremely short and essentially just presents a visual examination of animal prey caught by two individuals of the study species thirteen years ago. Even for a short communication this would seem very short. Is there any chance for captured items to have degraded in such a long period of storage? The introduction is sound, the methods are very brief (the collection process is not even described), the results are observational with no statistical methods, and the discussion mostly lists which organisms were observed in traps with some literature references for their behavior. It does not seem very international, although the authors say there are few studies previously which they could compare results to. I don't think this paper is sufficiently detailed for publication in PLOS One.

Reviewer #3: Dear authors,

This is a highly interesting study on the prey spectrum of a terrestrial carnivorous plant with suction traps. No data on prey spectrum in this species exists so far.

Here are some comments for the diverse secitions of this article:

For the M&M-section: please explain observation from the opened traps – did you wash out the trap content and investigated the medium washed out or did you only observed it in beneath the opened prey – did you use further aceto-ethanol for observation…. how was counting of so many prey-objects possible? Due to the similarities in sample handling – the article on the prey spectrum in aquatic U. gibba and U.inflata may be included, Gordon and Pacheco, 2007 (see: https://www.scielo.sa.cr/pdf/rbt/v55n3-4/art06v55n3-4.pdf ). This study also shows that, compared to vital investigations of trapped prey content like in Koller-Peroutka et al. 2015 – this approach with fixed traps in aceto-ethanol is also a powerful-tool for preservation of prey content and the results are comparable to the vital screening approach.

For the results section (and also for the discussion) some additional data would be favourable for the article: details on trap size and estimations on trap volume would be interesting for the readers – also to imagine how stuffed the traps maybe are. Furthermore, details on the trapping success for each single trap is important – are there differences in some traps? Did all traps catch successfully – or have been same traps with no prey? The numbers of prey objects for each prey are favourable for interpretation of the data. This is also interesting because of the limited sample size. Full data of individualised prey for each trap can also be included in the study or as a Table in a Supplement Part.

In the discussion section maybe this new article from March 2021 is of interest: Ceschin et al. in press, in ‘Plant Biosystems’ “Is the capture of invertebrate prey by the aquatic carnivorous plant Utricularia australis selective? (https://www.tandfonline.com/doi/full/10.1080/11263504.2021.1897704)

6. PLOS authors have the option to publish the peer review history of their article (what does this mean?). If published, this will include your full peer review and any attached files.

Reviewer #1: **Yes: **Lubomír Adamec

Reviewer #2: No

Reviewer #3: No

---

## [Author Response · Author response to Decision Letter 0]

24 Mar 2021

Dear Editor,

Thank you very much for handling our manuscript and for sending us the reviews. We appreciate the constructive feedback very much. Please find below our responses to the individual reviewers. We hope that our manuscript is now in a publishable format for PLoS ONE.

Kind regards,

Simon Poppinga (on behalf of the other authors).

Responses to reviewer 1

This interesting paper by German authors aims at determining the prey spectrum in traps of aquatic Utricularia multifida, an exotic and evolutionally early-diverging lineage of the species endemic to Australia. As there are disputations among different scientific teams concerning the functioning of U. multifida traps (opened passive vs. actively capturing closed traps), this novel study not only describes the field-based prey spectrum of this remarkable species for a comparative purpose with other aquatic Utricularia, but also indirectly determines the trap functioning and, moreover, sheds light on the phylogeny of Utricularia suction traps. In this study, the authors have convincingly proven that field-grown U. multifida does capture its prey of different taxonomical groups like other typical aquatic Utricularia species, but the mechanism of prey capturing stays unclear. I add some comments or questions to improve slightly the manuscript.

 Response: We thank reviewer 1 for the positive feedback.

p.3, l.58-59: “suction could not be observed in traps during laboratory experiments [29-31],….“ : maybe that it is not so important that they were laboratory experiments but that the studies (at least No. 31 – Plachno et al. 2019) were conducted on TC-raised plants. This fact could be crucial for the type of functioning of U. multifida traps: active in the field but passive in TC. I suggest that the authors mention that e.g.: “ observed in traps of tissue-culture raised plants during laboratory experiments [29-31],……“

Response: Since only one of the three references cited reports of experiments conducted on tissue-culture raised plant material, we decided to not apply the proposed change.

p.4, l.79: Could the authors specify what was the approximate size/length of the traps? Range?

Response: We added the following text to the Results section and included the respective SI Table: “With an average trap length of 2.3 mm (min: 1.8 mm; max: 2.7 mm) and width of 1.9 mm (min: 1.5 mm; max: 2.3 mm) (SI Table), we observed a similar overall size as reported in previous literature [27-32].”

p.5 or 7: Could the authors specify the maximal approximate length of the prey captured? This datum might be substantial for decision whether the traps are active (i.e., negative pressure, water suction) or passive (i.e., eel traps). In conclusion, this short communication should be published after a minor revision.

Response: We added the following text to the Results section: “Most captured prey had a size between 0.2-0.9 mm. Only a few prey items exceeded this length. The largest intact prey item we found was an ostracod with a body length of 1.2 mm. The total sizes of the captured coleopteran larva and midge larvae can be estimated to be around 1.3-1.8 mm, based on the sizes of the preserved head capsules found inside the traps.”

Responses to reviewer 2

This paper is interesting and contributes to an area of the literature which seems quite understudied, but it is extremely short and essentially just presents a visual examination of animal prey caught by two individuals of the study species thirteen years ago. Even for a short communication this would seem very short. Is there any chance for captured items to have degraded in such a long period of storage?

Response: We thank reviewer 2 for his/her efforts in reviewing our manuscript. We are convinced that no significant degradation had occurred, since the prey morphology and habit are conserved (while DNA might indeed have degraded somewhat over that time). The studied prey items are crustaceans, which have solid carapaces (insoluble in alcohol) which maintain all specific characters for taxonomic identification even in 200-year-old museum material (stored in alcohol or air dried). 13 years are comparatively short time spam for stored biological specimens.

The introduction is sound, the methods are very brief (the collection process is not even described)

Response: No additional info could be added to the description of Material and Methods we provide in the manuscript: two plants were randomly picked and traps detached from the plant and fixed in aceto-ethanol.

the results are observational with no statistical methods,and the discussion mostly lists which organisms were observed in traps with some literature references for their behavior.

Response: Since we were not able to collect and identify prey from different plants and/or habitats comparatively, statistical analyses are not necessary in our approach.

It does not seem very international, although the authors say there are few studies previously which they could compare results to. I don't think this paper is sufficiently detailed for publication in PLOS One.

Response: We strongly disagree with reviewer 2 in this point. Since the respective plant material is very difficult to obtain and no other published records are available, our results constitute the only available data on the diet of this enigmatic carnivorous plants. We are pleased to see that the editor as well as reviewers 1 and 3 share our opinion that the data presented are indeed important and robust, thereby warranting publication in a respectable peer-reviewed and international journal.

Responses to reviewer 3

Dear authors, This is a highly interesting study on the prey spectrum of a terrestrial carnivorous plant with suction traps. No data on prey spectrum in this species exists so far. Here are some comments for the diverse secitions of this article:

 Response: We thank reviewer 3 for the positive feedback.

For the M&M-section: please explain observation from the opened traps – did you wash out the trap content and investigated the medium washed out or did you only observed it in beneath the opened prey – did you use further aceto-ethanol for observation….

Response: We added the following text to the Materials and Methods section: “Traps were opened with forceps at the Department of Animal Ecology, Evolution and Biodiversity of the Ruhr-University Bochum, Germany and prey was carefully rinsed out. Block bowls were used and covered with a glass plate as often as possible to keep evaporation low (and thus evaporation-induced convection in the samples).”

how was counting of so many prey-objects possible?

Response: We added the following text to the Materials and Methods section: “Prey was separated from detritus in several portions, presorted into groups with the help of eyelashes, photographed with an Olympus SZX 16 (Olympus, Tokio, Japan) stereo microscope equipped with a TSO camera (Thalheim Spezialoptik GmbH, Pulsnitz, Germany), counted and identified according to the literature [34-41]. Trap sizes were measured with the same optical setup.”

Due to the similarities in sample handling – the article on the prey spectrum in aquatic U. gibba and U.inflata may be included, Gordon and Pacheco, 2007 (see: https://www.scielo.sa.cr/pdf/rbt/v55n3-4/art06v55n3-4.pdf ).

Response: We added the respective reference in the Materials and Methods section: “Plant material was fixed in aceto-ethanol (3:1) (cf. [9,23]).”

This study also shows that, compared to vital investigations of trapped prey content like in Koller-Peroutka et al. 2015 – this approach with fixed traps in aceto-ethanol is also a powerful-tool for preservation of prey content and the results are comparable to the vital screening approach.

Response: We added the respective reference in the Materials and Methods section: “Plant material was fixed in aceto-ethanol (3:1) (cf. [9,23]).”

For the results section (and also for the discussion) some additional data would be favourable for the article: details on trap size and estimations on trap volume would be interesting for the readers – also to imagine how stuffed the traps maybe are.

Response: We now give additional data, see our response to reviewer 1. However, only length and widths measurements are available and unfortunately no volumetric data could be obtained due to the complex architecture of the traps. In a future approach we are planning to measure trap volumes directly via modified µCT specimen preparation protocols, see our recently published paper: Westermeier et al. (2020) Annals of Botany 126: 1099–1107. DOI:10.1093/aob/mcaa135.

Furthermore, details on the trapping success for each single trap is important – are there differences in some traps? Did all traps catch successfully – or have been same traps with no prey? The numbers of prey objects for each prey are favourable for interpretation of the data. This is also interesting because of the limited sample size. Full data of individualised prey for each trap can also be included in the study or as a Table in a Supplement Part

Response: As we were initially aiming mostly for the prey spectrum of U. multifida and 17 traps were undoubtedly too few to examine the traps’ foraging characteristics, we (unfortunately) pooled the prey per plant, which prevents a trap-wise analysis. Accordingly, we cannot tell the maximum number of prey per trap, but even the averaged prey number with 13 animals per trap is impressive.

In the discussion section maybe this new article from March 2021 is of interest: Ceschin et al. in press, in ‘Plant Biosystems’ “Is the capture of invertebrate prey by the aquatic carnivorous plant Utricularia australis selective? (https://www.tandfonline.com/doi/full/10.1080/11263504.2021.1897704)

Response: We added the new reference [13] in the Discussion and changed the subsequent reference numbers accordingly: “Prey spectra have been thoroughly investigated for several free-floating aquatic species from U. section Utricularia, revealing that members … are commonly caught [4-13].”

Additional changes:

We added titles for Figure 1 and Table 1.

The acknowledgments are now more detailed.

---

## [Editor Report · Decision Letter 1]

29 Mar 2021

Snapshot prey spectrum analysis of the phylogenetically early-diverging carnivorous Utricularia multifida from U. section Polypompholyx (Lentibulariaceae)

PONE-D-21-00372R1

Dear Dr. Poppinga,

We’re pleased to inform you that your manuscript has been judged scientifically suitable for publication and will be formally accepted for publication once it meets all outstanding technical requirements.

Kind regards,

Ofer Ovadia

Academic Editor

PLOS ONE
---

## [Editor Report · Acceptance letter]

30 Mar 2021

PONE-D-21-00372R1 

Snapshot prey spectrum analysis of the phylogenetically early-diverging carnivorous *Utricularia multifida* from *U.* section *Polypompholyx* (Lentibulariaceae) 

Dear Dr. Poppinga:

I'm pleased to inform you that your manuscript has been deemed suitable for publication in PLOS ONE. Congratulations! Your manuscript is now with our production department. 

Kind regards, 

on behalf of

Dr. Ofer Ovadia 

Academic Editor

PLOS ONE